# Pre-exposure to mechanical ventilation and endotoxemia increases *Pseudomonas aeruginosa* growth in lung tissue during experimental porcine pneumonia

Jesper Sperber [1,2]*, Axel Nyberg[1,2], Anders Krifors[3,4], Paul Skorup[2], Miklós Lipcsey[5], Markus Castegren[4,6]

**1** Centre for Clinical Research Sörmland, Uppsala University, Uppsala, Sweden, **2** Department of Medical Sciences, Uppsala University, Uppsala, Sweden, **3** Centre for Clinical Research, Region of Västmanland, Uppsala University, Uppsala, Sweden, **4** Department of Physiology and Pharmacology, FyFa, Karolinska Institutet, Stockholm, Sweden, **5** Hedenstierna laboratory, CIRRUS, Anesthesiology and Intensive Care, Department of Surgical Sciences, Uppsala University, Uppsala, Sweden, **6** Perioperative Medicine and Intensive Care (PMI), Karolinska University Hospital, Stockholm, Sweden

* jesper.sperber@regionsormland.se

## Abstract

### Background

Immune system suppression during critical care contributes to the risk of acquired bacterial infections with *Pseudomonas (P.) aeruginosa*. Repeated exposure to endotoxin can attenuate systemic inflammatory cytokine responses. Mechanical ventilation affects the systemic inflammatory response to various stimuli.

### Aim

To study the effect of pre-exposure to mechanical ventilation with and without endotoxin-induced systemic inflammation on *P. aeruginosa* growth and wet-to-dry weight measurements on lung tissue and plasma and bronchoalveolar lavage levels of tumor necrosis factor alpha, interleukins 6 and 10.

### Methods

Two groups of pigs were exposed to mechanical ventilation for 24 hours before bacterial inoculation and six h of experimental pneumonia (total experimental time 30 h): $A_{30h+Etx}$ (n = 6, endotoxin 0.063 µg x kg$^{-1}$ x h$^{-1}$) and $B_{30h}$ (n = 6, saline). A third group, $C_{6h}$ (n = 8), started the experiment at the bacterial inoculation unexposed to endotoxin or mechanical ventilation (total experimental time 6 h). Bacterial inoculation was performed by tracheal instillation of 1x10$^{11}$ colony-forming units of *P. aeruginosa*. Bacterial cultures and wet-to-dry weight ratio analyses were done on lung tissue samples postmortem. Separate group comparisons were done between $A_{30h+Etx}$ vs.$B_{30h}$ (*Inflammation*) and $B_{30h}$ vs. $C_{6h}$ (*Ventilation Time*) during the bacterial phase of 6 h.

**Data Availability Statement:** All relevant data are within the paper and its Supporting Information files.

**Funding:** The RD funds of the Sörmland County Council and Uppsala University Hospital and the Family Olinder-Nielsen´s Foundation contributed with financial support to the main investigator JS. The funders had no role in study design, data collection and analysis, decision to publish, or preparation of the manuscript.

**Competing interests:** The authors have declared that no competing interests exist.

**Abbreviations:** AM, alveolar macrophage; ANOVA, analysis of variance; ARM, alveolar recruitment maneuver; BAL, bronchoalveolar lavage; CFU, colony forming unit; CI, cardiac index; *E.*, *Escherichia*; EDTA, ethylenediaminetetraacetic acid; ELISA, enzyme-linked immunosorbent assay; ET, endotoxin tolerance; Etx, endotoxin; $FiO_2$, inspired oxygen fraction; GLM, general linear model; h, hour; HR, heart rate; IL, interleukin; iNOS, inducible nitric oxide synthase; MAP, mean arterial pressure; min, minute; mmHg, millimeters of mercury; MV, mechanical ventilation; n, number; MPAP, mean pulmonary arterial pressure; $PaCO_2$, arterial partial pressure of carbon dioxide; $PaO_2$, arterial partial pressure of oxygen; PEEP, positive end-expiratory pressure; *P.*, *Pseudomonas*; P, pressure; PCWP, pulmonary capillary wedge pressure; RR, respiratory rate; s, second; SEM, standard error of the mean; TNF, tumor necrosis factor; $V_T$, tidal volume.

## Results

*P. aeruginosa* growth was highest in $A_{30h+Etx}$, and lowest in $C_{6h}$ (*Inflammation* and *Ventilation Time* both $p<0.05$). Lung wet-to-dry weight ratios were highest in $A_{30h+Etx}$ and lowest in $B_{30h}$ (*Inflammation* $p<0.01$, *Ventilation Time* $p<0.05$). $C_{6h}$ had the highest TNF-α levels in plasma (*Ventilation Time* $p<0.01$). No differences in bronchoalveolar lavage variables between the groups were observed.

## Conclusions

Mechanical ventilation and systemic inflammation before the onset of pneumonia increase the growth of *P. aeruginosa* in lung tissue. The attenuated growth of *P. aeruginosa* in the non-pre-exposed animals ($C_{6h}$) was associated with a higher systemic TNF-α production elicited from the bacterial challenge.

## Introduction

During critical illness, and especially during mechanical ventilation (MV), physical and immunological alterations occur that invite the high probability for infections with bacteria such as *Pseudomonas (P.) aeruginosa*. The attributable mortality from ventilator-associated pneumonia with *P. aeruginosa* is high. Moreover, treatment is increasingly difficult because of multidrug resistance and costs escalate from the prolonged length of stay [1]. The responsiveness and functionality of the individual immune system are central to the risk of developing an infection in general and to the onset of intensive care-related infections in particular [2]. Attempts have been made to describe the stage of reactivity of the immune system during sepsis based on clinical description or phenotype [3,4]. The ultimate goal is to tailor treatment to meet the clinical needs of patients based on the stage of immune reactivity. Concerning bacterial infections, any means to reduce inflammation-related organ damage while optimizing bacterial clearance is of utmost interest. We know from a previous experiment that twenty-four hours (h) of preceding MV combined with exposure to endotoxin attenuates the inflammatory cytokine response to a subsequent challenge of endotoxin, a phenomenon known as endotoxin tolerance (ET) [5]. Additionally, relatively small ventilatory interventions in healthy lungs, i.e. reduced tidal volumes from 10 to 6 mL x $kg^{-1}$ and elevated positive end-expiratory pressure (PEEP) from 5 to 10 $cmH_2O$, attenuate systemic and organ-specific inflammation and, most importantly, reduce *P. aeruginosa* burden in lung tissue in experimental pneumonia [6–8]. To our knowledge, scientific studies investigating weakened immune system responses to *in vivo* bacterial growth have not been performed in humans and are scarce in large animal models. As an experimental model animal, the size and anatomical features of the pig enable the use of machines and surveillance techniques relevant to intensive care. Additionally, the pig shares inflammatory traits with humans to the extent that it is suitable for pneumonia models [9]. The current experiment expands on the questions of inflammatory responsiveness and bacterial burden development after experimental intensive care. The underlying general hypothesis is that differences in inflammatory cytokine responsiveness, generated by any modality, correlate to differences in bacterial growth in lung tissue. Specifically, we hypothesized that two separate, clinically relevant entities, systemic inflammation and MV, would affect immune system reactivity and ultimately bacterial growth in experimental pneumonia. The primary aim was to investigate whether the bacterial growth in lung tissue six h after a bacterial challenge would be affected by a) a preceding inflammatory event, i.e. endotoxemia for 24 h and b) MV for 24 h.

These two separate parts of the current experiment are referred to as *Inflammation* (comparing two 30 h groups) and *Ventilation Time* (comparing one 30 h and one 6 h group). Secondary aims were to investigate the development of lung injury and cytokine responses in plasma and bronchoalveolar lavage (BAL).

## Materials and methods

The experiment contained three groups of healthy pigs of both sexes, between 9 and 12 weeks old and sexually immature (Fig 1). Two groups, $A_{30h+Etx}$ (n = 6) and $B_{30h}$ (n = 6), were studied for 30 h. The third group, $C_{6h}$ (n = 8), was derived from a previously published experiment and studied for 6 h [8]. Notably, the experiments were conducted simultaneously in time from the same animal batches. The animals were allocated to either experiment and experimental group by block randomization. The same methods and protocol were used for all groups. One animal in the 30-hour experiment and two in the six-hour experiment served as sham controls for a reference to normality and appreciation of the inflammatory model. The sham animals were not given endotoxin or a bacterial challenge but were in all other respects treated according to the protocol. A total of 24 animals were used in the production of the study. Additional information on methodology has been previously published [7] and a more detailed account is provided in the S2 File.

### Ethics statement

The animals were handled as per the regulations of the Swedish Board of Agriculture. The Animal Research Ethics Board of Uppsala approved and issued the permit for the current experiment (Uppsala djurförsöksetiska nämnd, DNr C 250/11). The study was designed in compliance with the Minimum Quality Threshold in Pre-clinical Sepsis Studies (MQTiPSS) guidelines and reported in adherence to the Animal Research: Reporting of In Vivo Experiments (ARRIVE) guidelines [10]. The animals (Swedish farm pig) were acquired from a private source, Mångsbo Gård breeding facility, Uppsala, Sweden. In total 23 animals were used in the current experiment. The animals were allowed to eat and drink *ad libitum* up to 1 h before the start of the experiment. The pigs were sedated with tiletamin 3 milligrams (mg) x kilogram (kg)-1, zolazepam 3 mg x kg-1, and xylacin 2.2 mg x kg-1. Morphine 20 mg and ketamine 100 mg were given in an auricular vein. Anesthesia was maintained with pentobarbital 8 mg x kg-1 x h-1 and morphine 0.26 mg x kg-1 x h-1. To facilitate ventilator management and counteract shivering and coughing muscle relaxation was maintained with an infusion of rocuronium at an initial rate of 2 mg x kg-1 x h-1.Immediately after the experimental

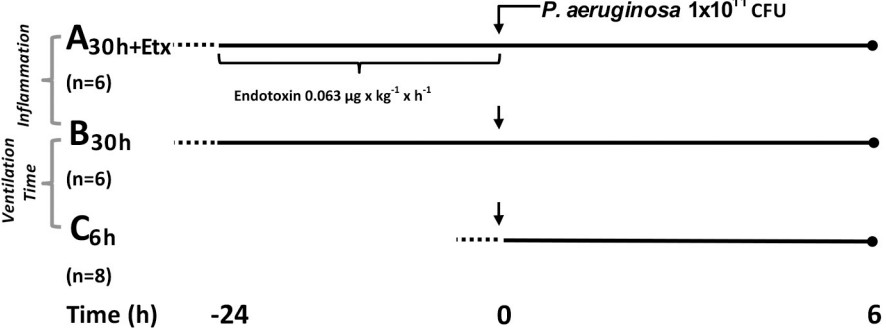

**Fig 1. Experimental overview.** Experimental parts *Inflammation* $A_{30h+Etx}$ vs. $B_{30h}$, *Ventilation Time* $B_{30h}$ vs. $C_{6h}$, Etx (endotoxin), dotted line indicates preparatory surgery of approximately 1 h, *Pseudomonas (P.) aeruginosa*, CFU (colony forming unit), the bacterial phase is between 0–6 h, n (number of animals per group).

endpoint, the animals were euthanized by an intravenous injection of potassium chloride and mechanical ventilation was withdrawn.

## Anesthesia and surgical procedure

Briefly, the anesthetized animals were tracheotomized and catheters were placed in a cervical artery, a central vein, the pulmonary artery, the portal vein by route of the splenic vein and the bladder.

## Protocol

The *P. aeruginosa* pneumonia phase made up the last 6 h in all groups and initial ventilator settings were the same in all groups. During the first 24 h in groups $A_{30h+Etx}$ and $B_{30h}$, the pigs were ventilated in a lateral position with 180˚ position changes every 6 h. During the last 6 h, the animals from all groups were in the supine position. During the preparatory surgery, a bolus of 20 mL x $kg^{-1}$ of Ringer´s acetate was administered. Group $A_{30h+Etx}$ received an endotoxin infusion of 0.063 µg x $kg^{-1}$ x $h^{-1}$ for 24 h in an auricular vein (*Escherichia (E.) coli*: 0111: B4 (Sigma Chemical Co., St Louis, MO, USA)) while group $B_{30h}$ received an equivalent amount of saline 0.9%. After 24 h, an intra-tracheal inoculum of $1x10^{11}$ colony forming units (CFU) of *P. aeruginosa* type O3 was given. Group $C_{6h}$ received no pre-exposure to ventilation but started the experimental protocol after preparatory surgery at the time of bacterial inoculation.

## Interventions

Initially, tidal volume was 10 mL x $kg^{-1}$, PEEP 5 cm $H_2O$, respiratory rate 25 x $min^{-1}$ and inspired oxygen fraction ($FiO_2$) 0.3. The respiration was adjusted to meet an arterial partial pressure of carbon dioxide ($PaCO_2$) value from 4.5–6.5 kilo pascal (kPa) by an increment or decrement in the respiratory frequency of 10%. Predefined increments of $FiO_2$ (0.3–0.6–0.8–1.0) were performed at arterial partial pressure of oxygen ($PaO_2$) values below 10 kPa simultaneously with changes in predefined PEEP levels (5-8-10-14 cm $H_2O$). If the plateau pressure were over 30 cm $H_2O$, the tidal volume was reduced to 7 mL x $kg^{-1}$ and the inspiratory to expiratory ratio was changed from 1:2 to 1:1. The alveolar recruitment maneuver (ARM) consisted of stepwise increments of PEEP (sequential 3 cm $H_2O$ increments for 5 seconds (s) each under control of systolic arterial blood pressure) until the peak pressure reached 35 cm $H_2O$, followed by prolonged inspiration for 10 s. ARM was performed at the start of the protocol (-24 h) and after each change of position in the 30 h groups, as well as after bacterial inoculation at 0 h in all groups. During the first 90 minutes (min) of the experiment, norepinephrine was used in boluses of 40 µg if mean arterial pressure (MAP) equaled mean pulmonary arterial pressure (MPAP). A MAP value, regardless of MPAP, that was below 60 mmHg after 90 min was treated with a bolus of Ringer´s acetate 15 mL x $kg^{-1}$, a 1 mL bolus of norepinephrine 20 µg x $mL^{-1}$, followed by a norepinephrine infusion of the same concentration starting at 5 mL x $h^{-1}$. At relapse of MAP below 60 mmHg, the infusion dose was doubled.

## Bacterial inoculum

The inoculation dose of *P. aeruginosa*, O-antigen serotyped to O3 by a slide agglutination test with commercial antisera (Bio-Rad Laboratories AB, Solna, Sweden) at the section for Clinical Microbiology and Infectious Medicine (Uppsala, Sweden), was a suspension of $1x10^{11}$ CFU dissolved in Lysogeny Broth according to Miller (VWR, Leuven, Belgium) at a total volume of 20 mL. The concentration of the suspension was estimated from an optical density reading

and afterwards verified from overnight cultures. Before the start of the study protocol at 0 h, a suction catheter was inserted blindly into the tracheal tube until it reached mechanical resistance. After a BAL was secured for culture and laboratory analyses the bacterial suspension was injected blindly into the lungs via the same catheter.

## Measurements and bacterial cultures

Physiological measurements and handling of laboratory samples followed earlier described sequences [7]. Six lung tissue samples were taken from each animal at the experimental endpoint. Three samples were from dorsal locations in the right upper, middle and lower lobes and three from the corresponding levels in the left lung (total of six samples). Bacterial cultures were performed after sequential dilution with saline 0.9% on Cystine-Lactose-Electrolyte-Deficient agar (BD Diagnostics, Stockholm, Sweden) plates kept at 37° C overnight. Six lung tissue samples of approximately 1 g were homogenized with 3 mL of saline 0.9% for 4 min with a Stomacher 80 Biomaster (Seward, Worthing, UK) before the single line sequential dilution. Six larger tissue samples of approximately 10 g were weighed before and after desiccation in 60° C for 12 h. BALs were performed directly before the start of the experiment at -24 h in the 30 h groups, in all groups at 0 h and before the experimental endpoint at 6 h. BAL was performed as a blind bronchial sampling with a suction catheter through the endotracheal tube with 20 mL of saline 0.9%. The BAL fluid was used for bacterial cultures after sequential dilution and cytokine measurements. Commercial porcine-specific sandwich enzyme-linked immunosorbent assay (ELISA) was used to determine tumor necrosis factor alpha (TNF-$\alpha$), interleukin (IL) 6 and IL10 in plasma and BAL (only TNF-$\alpha$ and IL6), (DY690B(TNF-$\alpha$) and DY686 (IL-6), R&D Systems, Minneapolis, MN, USA and KSC0102 (IL-10), Invitrogen, Camarillo, CA, USA). The lower detection limits in EDTA plasma were $<230$ pg $\times$ mL$^{-1}$ for TNF-$\alpha$, $<60$ pg $\times$ mL$^{-1}$ for IL6 and $<60$ pg $\times$ mL$^{-1}$ for IL10. All ELISAs had intra-assay coefficients of variation (CV) of $<5\%$ and a total CV of $<10\%$. After enzymatic conversion of nitrate to nitrite by nitrate reductase, total nitrite concentration in urine and BAL at -24, 0 and 6 h was measured using the Parameter™ assay (SKGE001, R&D Systems, Minneapolis, MN, USA). The urine samples were diluted 1:5 before the assay according to the recommendations of the manufacturer. Plasma and BAL urea (reagent: 7D75-21) and albumin (reagent: 7D54-21, BCP method) were measured on a Mindray BS380 chemistry analyzer (Mindray, Shenzhen, China) with the reagents obtained from Abbott laboratories (Abbott Park, IL, USA). The total coefficient of variations (CV) were 2% at 10 mmol/L for the urea method and 1% at 30 g/L for the albumin method.

## Statistics

The animals were allocated to groups of 6–8 animals by block randomization simultaneously with the previously published experiment [8]. Pro-inflammatory cytokine peaks during the first 24 h were anticipated in the two 30 h groups (A$_{30h+Etx}$ and B$_{30h}$). These differences in relation the 6 h group (C$_{6h}$)—with no pre-exposure before the bacterial challenge—constituted part of the constructed group separating inflammatory state characteristics. Therefore, comparative group statistics in the experimental parts *Inflammation* (A$_{30h+Etx}$ vs. B$_{30h}$) and *Ventilation Time* (B$_{30h}$ vs. C$_{6h}$) were based on data solely from the last 6 h of the experiment (the bacterial phase). No multigroup comparisons including all three groups were used in the experiment. A general linear model (GLM) was used for group comparisons in the lung tissue sample variables (i.e. bacterial growth and wet-to-dry weight ratio). Random effects were introduced into the model to account for the within-subject dependencies of the six simultaneous lung tissue samples from each individual, making the GLM a mixed model. Because the

bacterial inoculum was delivered blindly to either the right or left lung, the tissue samples in each animal were statistically analyzed using three levels (cranial-middle-caudal) consisting of the right and left corresponding samples. Raw data graphical presentations of the lung tissue sample variables related to side and cranio-caudal distributions are presented in the supplementary files, and *post hoc* comparative statistics was done by ANOVAs (S1 and S2 Figs). Repeated measures were analyzed with analysis of variance (ANOVA) for repeated measures. Only the group factor is presented as a p-value in the results from either the GLM or ANOVA for repeated measures. Inoculated dose and bacterial counts in BAL were analyzed with Mann-Whitney U-test for each experimental part (*Inflammation* and *Ventilation Time*) based on non-normal distribution, but data was presented in a logarithmic form for coherence within the presentation. In analogy with earlier publications all cytokines were logarithmically transformed [6–8]. Urinary nitrite was analyzed with Mann-Whitney U-test for each experimental part based on the mean concentration value from three measurements during the bacterial phase. Statistica$^{TM}$ (Statsoft, Tulsa, OK, version 13) was used for the statistical calculations and control of relevant assumptions. A p-value of $< 0.05$ was considered significant. A senior statistician approved the statistical design. Sham animals are only presented in the supplementary material as descriptive data. No power calculation was conducted for this specific experiment since we had no previous data on bacterial behavior in our models. Instead, we used the power calculation for the preceding inflammatory experiments [6–8]. It was based on a systemic TNF-alpha difference of 15% at 6 hours, an alpha error of 0,05, a power of 0,8, and an SD of 10%, which yielded six evaluable animals per group. The choice of 8 animals per group in the previously published day-based experiment [8] was based on this calculation while allowing for a slightly larger variability in the bacterial outcome variable. As we started with the day-based experiment we could appreciate the bacterial growth in lung tissue better. Based on this data we reduced the number of animals in the 30 h experiments, which were completed at the end of the experimental period, from eight to six to meet the 3R principle. In summary, we reduced the number of animals as we believed we could meet the required difference in the main outcome variable anyway.

## Results

No animal died before the experimental endpoint at 6 h after the bacterial challenge. The animals had a mean weight of $25.2 \pm 2$ kg with no differences between the groups. The use of norepinephrine and fluid boluses in accordance with the protocol is described in Table 1. The larger dose of norepinephrine in group $B_{30h}$ could be traced to one animal that was circulatory unstable throughout the experiment. No differences were noted in the inoculation dose between any groups (Table 1). Regarding distribution within the lungs of bacterial growth and wet-to-dry ratios there was no difference between right and left sided samples. There was a significant difference in cranio-caudal direction regarding bacterial growth. Raw data of all samples in the experiment is presented in S1 and S2 Figs.

### Experimental part *Inflammation*–$A_{30h+Etx}$ compares to $B_{30h}$

**$Pseudomonas\ aeruginosa$ in lung tissue and bronchoalveolar lavages.** The highest bacterial growth was in group $A_{30h+Etx}$ (Fig 2). No differences were detected in bacterial growth between the groups in BAL (Table 2).

**Physiologic, hypoperfusion and lung injury variables.** We found no differences between the groups in the ventilator variables; peak- (P peak), plateau- (P plateau), mean airway pressure (P mean), respiratory rate (RR) and tidal volume ($V_T$); nor were there any differences in the circulatory variables: heart rate (HR), MPAP, pulmonary capillary wedge pressure

**Table 1. Norepinephrine, fluid, inoculation dose.**

| Variable | Group | a) Pre-exposure | b) Bacterial phase | p |
|---|---|---|---|---|
| **Norepinephrine** | $A_{30h+Etx}$ | 150(0/400) | 200(0/600) | |
| (µg) | $B_{30h}$ | 100(0/600) | 300(0/1200) | 0.94 |
| | $C_{6h}$ | - | 0(0/40) | 0.28 |
| **Fluid bolus** | $A_{30h+Etx}$ | 1 | 0 | |
| (n) | $B_{30h}$ | 1 | 0 | N/A |
| | $C_{6h}$ | - | 0 | N/A |
| **Inoculation dose** | $A_{30h+Etx}$ | - | 11.0(10.9/11.1) | |
| ($\log_{10}$ CFU) | $B_{30h}$ | - | 10.8(10.8/11.1) | 0.60 |
| | $C_{6h}$ | - | 10.9(10.7/11.0) | 0.52 |

Norepinephrine and fluid boluses expressed in separated parts as a) dose during the first 24 h in $A_{30h+Etx}$ and $B_{30}$, and b) dose during the bacterial phase 0–6 h, (3/6 animals in $A_{30h+Etx}$ and $B_{30}$ each and 2/8 in $C_6$ received norepinephrine), fluid boluses of Ringer´s acetate 15 mL x kg$^{-1}$ expressed in absolute numbers, colony forming unit (CFU), inoculation dose given at 0 h, median(LQ/HQ), Mann-Whitney U-test on the total amounts during the experiment, p upper refers to $A_{30h+Etx}$ vs. $B_{30h}$, lower to $B_{30h}$ vs. $C_{6h}$, N/A (not applicable).

(PCWP), MAP, cardiac index (CI) and lactate. Moreover, we found no differences in the arterial partial pressure of oxygen to the inspired oxygen fraction ratio ($PaO_2$/$FiO_2$) (Table 3). The wet-to-dry weight ratio was lower in $B_{30h}$ than in $A_{30h+Etx}$ (Fig 3). Additional analyzes of albumin in plasma and BAL were performed to calculate the alveolo-capillary permeability. However, too many albumin values were below detection limit in BAL to make the calculation. The data for albumin is presented in S1 Data.

**Inflammatory variables in plasma, bronchoalveolar lavages and urine.** The plasma levels of TNF-α displayed early peak values during the pre-exposure in $A_{30h+Etx}$, but there were no differences to $B_{30h}$ in the last 6 h (Fig 4). IL6 showed biphasic peak values, during the pre-exposure and at the end of the experiment, that did not separate the groups (Table 4 and S3 Fig). No differences were seen in IL10, leukocytes, neutrophils, temperature or platelets between the groups (Table 4). Finally, we found no differences between the groups in TNF-α

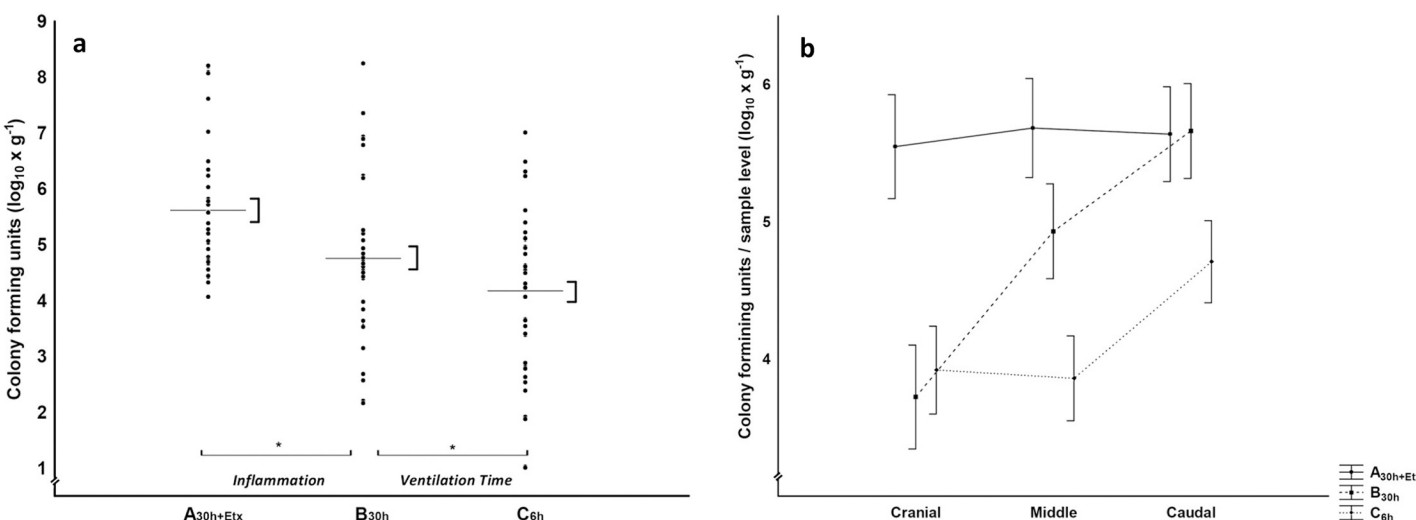

**Fig 2. Bacterial growth in lung tissue.** *P. aeruginosa* counts, $\log_{10}$ colony forming units per gram lung tissue. a) Dots represent raw data for each group, mean±SEM (line, brackets), general linear model, * denotes p<0.05, experimental parts *Inflammation* $A_{30h+Etx}$ vs. $B_{30h}$, *Ventilation Time* $B_{30h}$ vs. $C_{6h}$. b) Descriptive graphical presentation of distribution based on sample level (cranial, middle, caudal) for each group connected by a line, brackets SEM for each level.

**Table 2. *Pseudomonas aeruginosa* and cytokines in bronchoalveolar lavages.**

| Variable | Group | -24 h | 0 h | 6 h | p |
|---|---|---|---|---|---|
| *P. aeruginosa* BAL | $A_{30h+Etx}$ | 0.0(0.0/0.0) | 0.0(0.0/0.0) | 4.0(3.8/4.0) | |
| ($\log_{10}$CFU x $100\mu L^{-1}$) | $B_{30h}$ | 0.0(0.0/0.0) | 0.0(0.0/0.0) | 4.4(4.1/5.0) | 0.24 |
| | $C_{6h}$ | | 0.0(0.0/0.0) | 3.8(3.3/4.6) | 0.40 |
| **TNF-α BAL** | $A_{30h+Etx}$ | 1.6(1.0/2.3) | 2.2(2.1/3.0) | 3.5(2.6/3.5) | |
| ($\log_{10}$ng x $L^{-1}$) | $B_{30h}$ | 1.5(1.2/1.7) | 1.8(1.6/3.0) | 3.4(2.7/3.5) | 0.93 |
| | $C_{6h}$ | | 1.8(1.7/1.9) | 3.3(2.8/3.6) | 0.75 |
| **IL6 BAL** | $A_{30h+Etx}$ | 2.8(1.7/2.9) | 2.1(1.7/2.9) | 2.3(1.7/2.9) | |
| ($\log_{10}$ng x $L^{-1}$) | $B_{30h}$ | 2.1(2.0/2.6) | 2.9(2.7/3.1) | 2.8(2.3/3.1) | 0.58 |
| | $C_{6h}$ | | 2.6(1.7/2.9) | 2.6(2.0/3.0) | 0.65 |

*Pseudomonas* (*P.*) *aeruginosa*, bronchoalveolar lavage (BAL), colony forming unit (CFU), TNF-α (tumor necrosis factor alpha), IL6 (interleukin 6) at the start of the experiment -24 h ($A_{30h+Etx}$ and $B_{30h}$), at the bacterial inoculation 0 h and at the end of the experiment 6 h, median(LQ/HQ), p upper refers to $A_{30h+Etx}$ vs. $B_{30h}$, lower to $B_{30h}$ vs. $C_{6h}$ solely at 6h (no statistical differences between compared groups at -24 or 0 h presented in S1 Table), Mann-Whitney U test, N/A (not applicable).

or IL6 levels in BAL at the end of the experiment (Table 2). Additional analyzes of urea in plasma and BAL were performed to calculate the dilution factor in the BAL samples. However, too many urea values were below detection limit in BAL to make the calculation. The data for urea is presented in S1 Data. At the bacterial inoculation the nitrite concentration in urine was reduced in both groups to around 20% of values at the start of the experiment at -24 h. The mean concentration during the bacterial phase of the experiment, 0 to 6 h, did not separate the groups (Fig 5).

## Experimental part *Ventilation Time*–$B_{30h}$ compares to $C_{6h}$

*Pseudomonas aeruginosa* in lung tissue and bronchoalveolar lavages.   The higher bacterial growth was in $B_{30h}$ (Fig 2). There were no differences between the groups in BAL (Table 2).

**Physiologic, hypoperfusion and lung injury variables.**   For the ventilator variables (P peak, P plateau, P mean, RR) there were no differences between the groups. $V_T$ was higher in $C_{6h}$ than in $B_{30h}$. In the circulatory variables (HR, MPAP, PCWP) no differences were detected between the groups. $C_{6h}$ displayed lower MAP and CI and higher lactate than $B_{30h}$. $PaO_2/FiO_2$ was higher in $C_{6h}$ than in $B_{30h}$ (Table 3). The wet-to-dry weight ratio was lower in $B_{30h}$ than in $C_{6h}$ (Fig 3).

**Inflammatory variables in plasma and bronchoalveolar lavages.**   The TNF-α levels were higher in $C_{6h}$ than in $B_{30h}$ in plasma (Fig 4). No differences in IL6 (Table 4 and S3 Fig), IL10, leukocytes, neutrophils or temperature were seen between groups. Platelet levels were higher in $C_{6h}$ than in $B_{30h}$ (Table 4). At the end of the experiment, we observed no differences between the groups in TNF-α or IL6 in BAL (Table 2).

**Sham animals.**   For the sham animals (30 h, n = 1 and 6 h, n = 2), all variables are presented in the supplementary material as descriptive statistics (S2 Table).

## Discussion

Of the groups, the control group ($C_{6h}$) had the lowest bacterial growth, the strongest pro-inflammatory cytokine response to the bacterial stimulus and intermediate edema development. The two separate parts of the experiment are discussed in sequence, although the rationale underlying the bacterial outcomes is overlapping.

**Table 3. Physiologic variables, hypoperfusion and lung injury.**

| Variable | Group | -24 h | 0 h | 3 h | 6 h | p |
|---|---|---|---|---|---|---|
| **P peak** | $A_{30h+Etx}$ | 18±3 | 18±3 | 26±8 | 27±6 | |
| (cmH$_2$O) | $B_{30h}$ | 16±2 | 18±3 | 23±7 | 21±4 | 0.44 |
| | $C_{6h}$ | | 18±2 | 20±2 | 21±2 | 0.44 |
| **P plateau** | $A_{30h+Etx}$ | 16±4 | 16±3 | 23±7 | 26±6 | |
| (cmH$_2$O) | $B_{30h}$ | 15±2 | 17±3 | 20±5 | 20±4 | 0.32 |
| | $C_{6h}$ | | 17±2 | 20±2 | 21±2 | 0.27 |
| **P mean** | $A_{30h+Etx}$ | 9±2 | 9±1 | 11±3 | 13±4 | |
| (cmH$_2$O) | $B_{30h}$ | 9±2 | 9±1 | 10±2 | 10±2 | 0.35 |
| | $C_{6h}$ | | 9±1 | 9±1 | 9±1 | 0.06 |
| **Respiratory rate** | $A_{30h+Etx}$ | 25±0 | 25±0 | 28±6 | 31±10 | |
| (breath x min$^{-1}$) | $B_{30h}$ | 25±0 | 25±0 | 26±1 | 26±2 | 0.27 |
| | $C_{6h}$ | | 24±2 | 25±2 | 25±2 | 0.13 |
| **Tidal volume** | $A_{30h+Etx}$ | 239±54 | 210±37 | 185±46 | 182±50 | |
| (mL) | $B_{30h}$ | 224±22 | 216±33 | 199±33 | 189±21 | 0.67 |
| | $C_{6h}$ | | 248±17 | 251±21 | 245±19 | <0.01* |
| **HR** | $A_{30h+Etx}$ | 93±23 | 95±21 | 121±32 | 137±45 | |
| (beats x min$^{-1}$) | $B_{30h}$ | 93±19 | 94±32 | 121±27 | 126±21 | 0.82 |
| | $C_{6h}$ | | 103±20 | 94±9 | 111±12 | 0.12 |
| **MPAP** | $A_{30h+Etx}$ | 18±3 | 18±3 | 25±7 | 26±4 | |
| (mmHg) | $B_{30h}$ | 18±1 | 17±2 | 25±5 | 24±5 | 0.93 |
| | $C_{6h}$ | | 22±4 | 29±14 | 34±14 | 0.31 |
| **PCWP** | $A_{30h+Etx}$ | 8±2 | 6±3 | 8±3 | 7±2 | |
| (mmHg) | $B_{30h}$ | 8±2 | 5±2 | 7±2 | 8±1 | 0.7 |
| | $C_{6h}$ | | 9±3 | 9±4 | 9±4 | 0.35 |
| **CI** | $A_{30h+Etx}$ | 3.1±1.0 | 3.2±0.8 | 3.9±1.3 | 4.2±1.7 | |
| (L x min$^{-1}$ x m$^{-2}$) | $B_{30h}$ | 2.7±0.5 | 3.6±1.0 | 3.9±1.1 | 3.7±1.0 | 0.53 |
| | $C_{6h}$ | | 3.2±0.7 | 2.5±0.3 | 2.4±0.8 | <0.05* |
| **MAP** | $A_{30h+Etx}$ | 88±14 | 81±5 | 99±8 | 97±22 | |
| (mmHg) | $B_{30h}$ | 83±15 | 75±10 | 88±10 | 88±15 | 0.14 |
| | $C_{6h}$ | | 93±22 | 79±15 | 75±20 | <0.05* |
| **Lactate** | $A_{30h+Etx}$ | 1.9±0.9 | 0.9±0.3 | 1.1±0.3 | 1.1±0.3 | |
| (mmol x L$^{-1}$) | $B_{30h}$ | 1.8±0.3 | 1.1±0.2 | 1.4±0.3 | 1.4±0.5 | 0.18 |
| | $C_{6h}$ | | 1.8±0.4 | 1.6±0.7 | 1.5±0.6 | <0.05* |
| **PaO$_2$/FiO$_2$** | $A_{30h+Etx}$ | 450±44 | 348±127 | 244±133 | 213±125 | |
| | $B_{30h}$ | 489±38 | 339±92 | 270±149 | 232±147 | 0.86 |
| | $C_{6h}$ | | 446±43 | 378±56 | 343±61 | <0.05* |

Pressure (P), heart rate (HR), mean pulmonary arterial pressure (MPAP), pulmonary capillary wedge pressure (PCWP), cardiac index (CI), mean arterial pressure (MAP), arterial partial pressure of oxygen (PaO$_2$), inspired oxygen fraction (FiO$_2$), mean±SD, p upper refers to $A_{30h+Etx}$ vs. $B_{30h}$, lower to $B_{30h}$ vs. $C_{6h}$, ANOVA for repeated measures

* denotes $p < 0.05$.

## Experimental part *Inflammation*–$A_{30h+Etx}$ compares to $B_{30h}$

$A_{30h+Etx}$ differed from $B_{30h}$ by higher bacterial counts in lung tissue and more evident edema formation, but there was no difference in systemic cytokine expression following the bacterial challenge. Group $A_{30h+Etx}$ was exposed to endotoxin for 24 h to model a state of general inflammation before the bacterial challenge. Both groups exhibited moderate IL6 peaks during

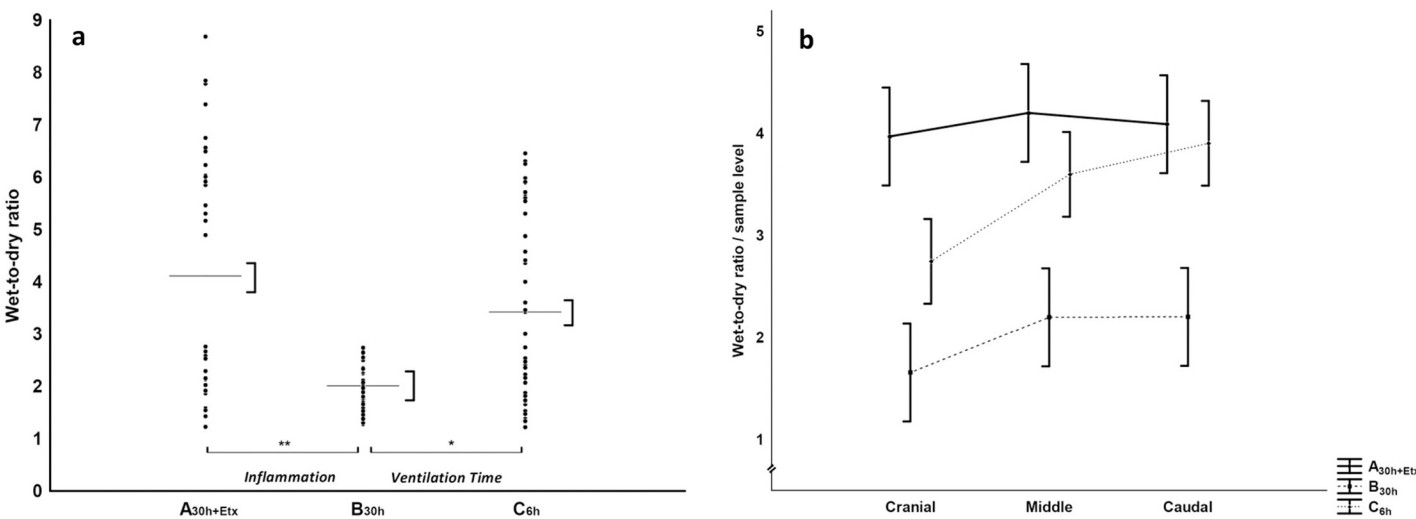

**Fig 3. Wet-to-dry weight ratio of lung tissue.** Wet-to-dry ratios of lung tissue. a) Dots represent raw data for each group, mean±SEM (line, brackets), general linear model, experimental parts *Inflammation* $A_{30h+Etx}$ vs. $B_{30h}$, *Ventilation Time* $B_{30h}$ vs. $C_{6h}$, * denotes p<0.05, ** denotes p<0.001. b) Descriptive graphical presentation of distribution based on sample level (cranial, middle, caudal) for each group connected by a line, brackets SEM for each level.

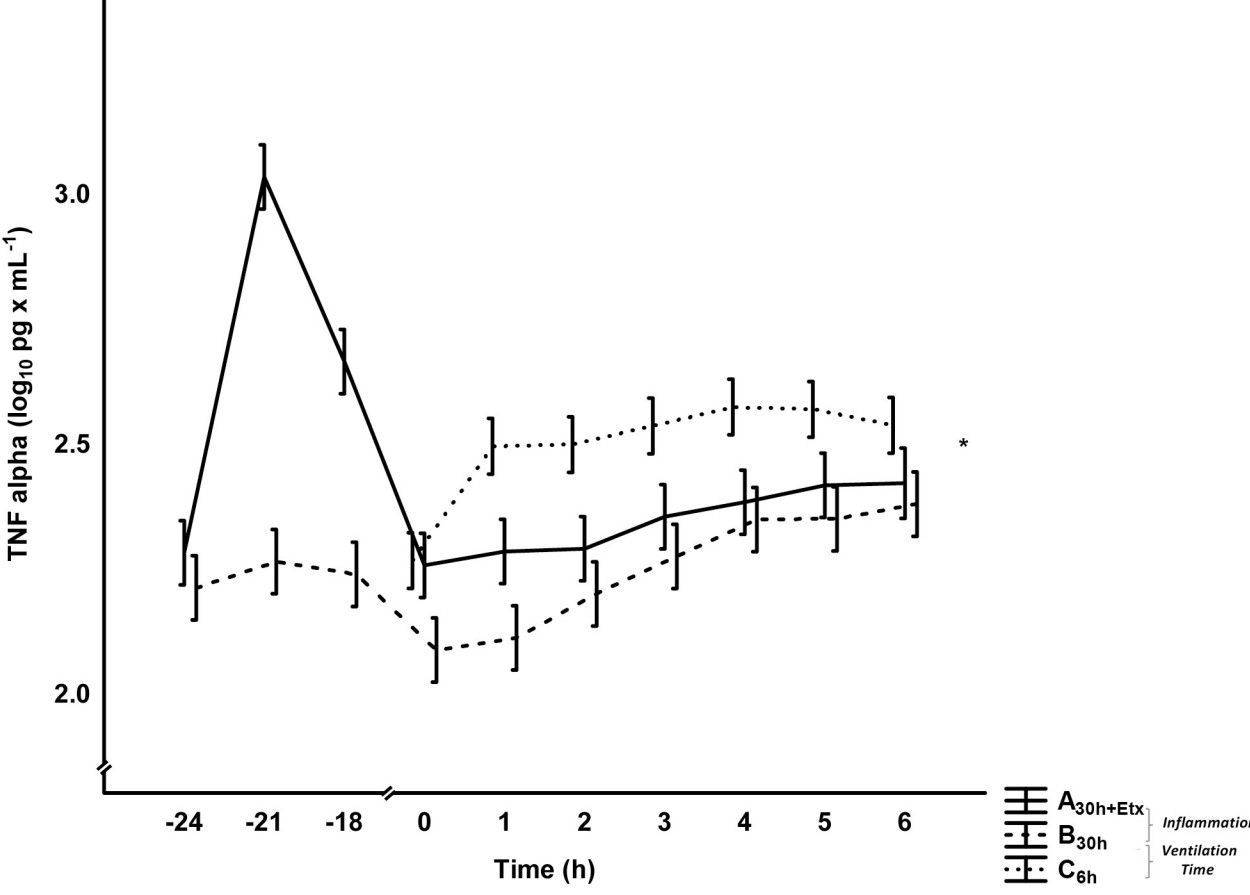

**Fig 4. TNF-α in plasma.** Mean±SEM, total group difference calculated between 0–6 h with ANOVA for repeated measures, experimental parts *Inflammation* $A_{30h+Etx}$ vs. $B_{30h}$, *Ventilation Time* $B_{30h}$ vs. $C_{6h}$, * denotes p<0.01 in *Ventilation Time*, axis scale changes at 0 hours.

**Table 4. Plasma cytokines, inflammatory cells and temperature.**

| Variable | Group | -24 h | 0 h | 3 h | 6 h | p |
|---|---|---|---|---|---|---|
| **IL6** | $A_{30h+Etx}$ | 1.9±0.6 | 1.9±0.6 | 2.4±0.1 | 2.7±0.3 | |
| ($\log_{10}$ ngxL$^{-1}$) | $B_{30h}$ | 1.9±0.6 | 2.1±0.3 | 2.5±0.3 | 2.7±0.5 | 0.84 |
| | $C_{6h}$ | | 1.1±0.3 | 2.4±0.3 | 3.1±0.3 | 0.63 |
| **IL10** | $A_{30h+Etx}$ | 1.0±0.9 | 0.4±0.6 | 0.3±0.6 | 0.9±0.6 | |
| ($\log_{10}$ ngxL$^{-1}$) | $B_{30h}$ | 0.3±0.7 | 0.6±0.7 | 0.3±0.6 | 1.0±1.0 | 0.99 |
| | $C_{6h}$ | | 0.8±0.6 | 0.7±0.6 | 0.8±0.7 | 0.40 |
| **Leukocytes** | $A_{30h+Etx}$ | 13±6 | 15±6 | 16±7 | 23±11 | |
| ($10^9$ x L$^{-1}$) | $B_{30h}$ | 16±6 | 19±7 | 18±6 | 21±7 | 0.99 |
| | $C_{6h}$ | | 14±5 | 15±6 | 18±7 | 0.38 |
| **Neutrophils** | $A_{30h+Etx}$ | 6±5 | 5±2 | 8±4 | 16±9 | |
| ($10^9$ x L$^{-1}$) | $B_{30h}$ | 8±5 | 10±8 | 11±6 | 15±6 | 0.61 |
| | $C_{6h}$ | | 7±3 | 9±4 | 11±6 | 0.61 |
| **Temperature** | $A_{30h+Etx}$ | 38.1±0.4 | 38.8±2.1 | 38.8±2.4 | 39.1±2.1 | |
| (°C) | $B_{30h}$ | 37.6±0.7 | 39.7±2.7 | 39.4±2.8 | 39.7±3.1 | 0.65 |
| | $C_{6h}$ | | 38.4±0.9 | 38.6±0.9 | 39.1±1.0 | 0.86 |
| **Platelets** | $A_{30h+Etx}$ | 385±162 | 209±128 | 200±118 | 200±111 | |
| ($10^9$ x L$^{-1}$) | $B_{30h}$ | 323±60 | 218±101 | 199±91 | 205±91 | 0.99 |
| | $C_{6h}$ | | 377±190 | 390±165 | 380±189 | <0.05* |

Interleukin (IL), mean±SD, p upper refers to $A_{30h+Etx}$ vs. $B_{30h}$, lower to $B_{30h}$ vs. $C_{6h}$, ANOVA for repeated measures

* denotes p<0.05.

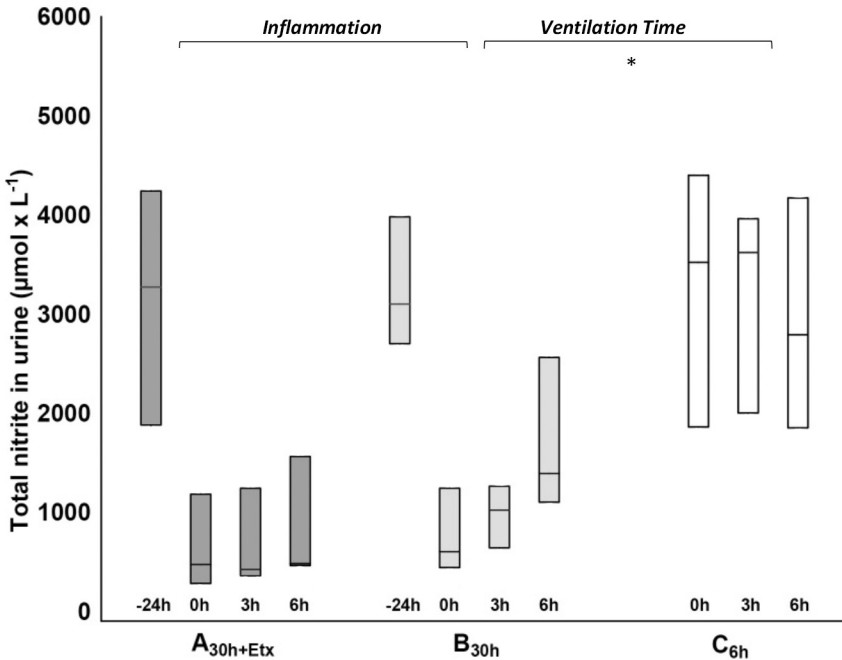

**Fig 5. Total nitrite in urine.** Total nitrite concentration (after conversion of nitrate to nitrite) in urine at -24 h in groups $A_{30h+Etx}$ (dark grey) and $B_{30h}$ (light grey), and in all groups during the bacterial phase 0, 3 and 6 h. Experimental part *Inflammation* $A_{30h+Etx}$ vs. $B_{30h}$ (p 0.30) and *Ventilation Time* $B_{30h}$ vs. $C_{6h}$ (white),* denotes p<0.05 in *Ventilation Time*. Mann-Whitney from mean concentration value in the bacterial phase 0–6 h. Median (HQ/LQ).

the first 24 h, but the early TNF-α reaction in $A_{30h+Etx}$ was considerably more pronounced than in $B_{30h}$. The difference in TNF-α response during the pre-exposure potentially affected the bacterial outcome in several ways. Endotoxin tolerance (ET) was originally described as a blunted fever response to a repeated endotoxemic challenge but has since proved to have multiple effects on the innate and the adaptive immune system toward a general state of reduced responsiveness [11]. The phenomenon is not limited to TNF-α expression but involves multiple cytokines to varying degrees. ET is believed to protect the host from excessive inflammatory reactions but also to bring an increased vulnerability to infections, indicating effects on the cellular defenses [12]. In a porcine experimental sepsis model using intra-venous *P. aeruginosa* the alveolar macrophages functionality was severely impaired [13]. In murine experiments ET was associated with increased survival in sepsis [14] and reduced *P. aeruginosa* clearance in lung tissue [15,16]. The relatively mild reactions in the pro-inflammatory cytokines, TNF-α and IL6, in $A_{30h+Etx}$ to the bacterial challenge were on these grounds anticipated in the present experiment, whereas the similar reaction in $B_{30h}$ was not. However, any non-infectious process that elicits a systemic inflammatory response, such as cardiac arrest or trauma, can potentially produce the same reduced cytokine response as endotoxin which could explain the mild cytokine reaction also in $B_{30h}$. Summarily, the early TNF-α surge in $A_{30h+Etx}$ from endotoxin likely produced a wider array of effects on bacterial growth than could be differentiated from the cytokine responses to the bacterial challenge. Cytokines have pleiotropic effects. For example, the magnitude of pro-inflammation of IL6 or anti-inflammation of IL10 is dependent on the timing of the insult, the type of target cell and any pre-exposure to other cytokines or inflammatory events [17]. The timing of an inflammatory cytokine surge has a determining influence on the host's handling of a *P. aeruginosa* challenge. IL4 given 24 h before a challenge in mice reduced both the TNF-α response and lethality. Conversely, simultaneous delivery has led to the opposite effects with enhanced TNF-α and lethality [14]. In another mouse experiment pre-treatment with anti-inflammatory IL10 reduced lung injury and lethality from intra-tracheal *P. aeruginosa* inoculation, indicating a possible role in bacterial outcome from the balance between pro- and anti-inflammatory cytokines at the time of bacterial challenge [18]. In the current experiment we did not find any significant differences between groups for IL10 or from *post hoc* comparisons during the bacterial phase as to the ratios of TNF-α/IL10 or IL6/IL10. Thus, other factors than cytokine levels or their respective ratios during the bacterial phase were more capable of exerting an effect on the bacterial outcome of *P. aeruginosa*.

Given no cardiac failure, edema in lung tissue is indicative of an inflammatory process and has long been associated with bacterial infection [19]. The presence of edema proved to have a negative effect on bacterial clearance in early experimental lung models and correlated with vulnerability to bacterial pneumonia [20]. TNF-α can initiate inflammatory edema in experimental settings [21] and sepsis and general inflammatory states affect the endothelial surfaces throughout the body [22]. Endotoxin in circulation results in shed glycocalyx from the endothelial luminal side of capillaries, thereby promoting fluid leak and edema formation. In human experimental settings this effect can be averted with a TNF-α blocking agent, indicating the direct effect of cytokine surges [23]. From this, we can hypothesize that the presence or escalation of edema during the bacterial phase in the last 6 h of our experiment resulted in impairing the immune system's ability to reduce bacterial growth. There is a likely effect from endotoxin exposure on cellular defenses in the present experiment. The alveolar macrophage (AM) has been characterized as central to bacterial clearance and is susceptible to inflammatory influence on functionality [24,25]. In several lung damage models, from MV with large $V_T$ and zero PEEP to *E. coli* endotoxin, the resulting inflammation suppressed AM phagocytic and bactericidal action [26–28]. Even severe infections outside the lungs can affect bacterial

clearance in the lungs [13]. Experimental *E. coli* peritonitis, releasing endotoxin and producing TNF-α in the liver, resulted in decreased alveolar neutrophil recruitment, phagocytosis and superoxide production in the lungs [15]. The same result was produced by intravenous endotoxin alone, highlighting the central role of cytokines in eliciting effects from different primary stimuli. The cellular defense functions of the AM are, to a variable degree, dependent on extracellular opsonization molecules, such as surfactant proteins from alveolar type II cells. Bacterial endotoxin is one of several agents that has multiple negative effects on surfactant production and function [29]. These changes, especially in surfactant protein-A and -D, lead to lower phagocytic and bactericidal ability from the AM [30]. Additionally, a condition of reduced surfactant production and function leads to atelectasis formation which, on its own, has a negative effect on bacterial clearance in porcine experimental settings [31]. Another point of influence of endotoxin is the complement system. A functional complement system is essential for an effective neutrophil function against Gram-negative bacteria, such as *P. aeruginosa* [32]. Endotoxin can deactivate complement by forming immune complexes with natural antibodies and so reduce functional levels [33]. Quantitative reductions in complement factor levels or functionality have, in experimental settings, severely hampered the antibacterial defenses of the lung [34]. Although these co-factors for effective cellular handling of bacteria, i.e. surfactant and complement, were not specifically addressed in this experiment, they remain likely contributors to the bacterial outcome.

As an inflammatory adjunctive measure and indication of inflammatory cell activity, we analyzed nitrate concentration in urine to proxy the total nitric oxide turnover during the experimental bacterial phase. Nitric oxide (NO) is an important inflammatory mediator molecule excreted from macrophages, neutrophils and endothelia upon inflammatory stimulus. The dominant part of NO production during an infection is attributed to inducible nitric oxide synthase (iNOS). Bursts of NO in response to inflammatory stimuli have multiple actions affecting direct bactericidal capacity, vascular permeability and scavenging of oxygen free radicals. The resulting effect from NO can differ depending on NO levels and timing during an infection and where NO exerts its dominant effect–in the mitochondrion of the macrophage or as part of the nitrous radical burst aimed at invading pathogens [35]. As the NO molecule is highly unstable measurements are practically directed toward stable metabolites such as nitrite, which is a significant source of NO in the system [36]. In the *Inflammation* experiment both groups had similarly suppressed nitrite levels in urine to around 20% of base line values after 24 h. The levels did not increase significantly during the bacterial phase in either group (Fig 5). A previous ovine endotoxin tolerance experiment has provided evidence of suppressed nitrite levels after endotoxin exposure and synchronously suppressed NOS activity harmonizing with the picture in the endotoxin exposed animals in group $A_{30h+Etx}$ in the current experiment [37]. The similar nitrite reduction in $B_{30h}$ was unexpected but speaks for a general suppression of inflammatory reactivity from unknown causes mentioned earlier.

Only small, non-significant differences were noted in inflammatory cells, hypoperfusion and physiologic variables. Most remarkable, given the large difference in edema formation, was the lack of difference in $PaO_2/FiO_2$. This result indicates a major edema development in $A_{30h+Etx}$ only after the bacterial challenge.

## Experimental part *Ventilation Time*–$B_{30h}$ compares to $C_{6h}$

$B_{30h}$ differed from $C_{6h}$ by higher bacterial counts, lower edema formation and lower TNF-α expression from the bacterial challenge. Group $B_{30h}$ was exposed to 24 h of anesthesia and MV before the bacterial challenge as a way to evaluate the potential impact of time in basic intensive care treatment before the bacterial challenge. The 24 h in MV before the bacterial

challenge in $B_{30h}$ resulted in differences in the inflammatory response, as well as the physiological state between the groups already at the time of bacterial challenge. These facts provide several plausible explanations for the bacterial outcome.

Group $C_{6h}$ reacted to the bacterial challenge with the full force of an unaffected innate immune system. The unaffected reaction led to an extensive inflammatory reaction with pro-inflammatory cytokines and edema development in lung tissue. Mortality in the early stages of sepsis is often related to profound inflammatory reactions, whereas later mortality relates to multiple organ failure or unresolved infections [38,39]. In $C_{6h}$ there was no previous inflammatory event that could induce ET and no previous cytokine surge that could affect macrophage function. Therefore, the higher TNF-α response in plasma to the bacterial challenge in $C_{6h}$ is an indication of the preserved functionality of the alveolar macrophages, as they are the most considerable source of cytokines in the inflammatory response from the lungs [40]. Additionally, urinary nitrate levels were significantly higher in the unexposed animals of $C_{6h}$ in comparison to the ventilated animals of $B_{30h}$. The higher nitrite levels in $C_{6h}$ speaks for a higher NO activity from macrophages, neutrophils and endothelia and is a possible mechanistic explanation for both the lower bacterial levels and the greater edema development.

MV can produce inflammation if performed carelessly with cyclic atelectasis and over-stretched alveoli, but in recent years the concept of protective ventilation has become the standard [41]. The term protective refers in the broader sense to the avoidance of inflicting iatrogenic harm. In our experiment the respiratory settings were moderate for PEEP and $V_T$, and intermittent recruitments were done to avoid inflammation originating from the ventilator. However, during a general inflammatory stimulus supposedly safe ventilator settings can turn harmful and produce lung injury [42]. The development of lung injury is strongly correlated to immune system reactions with neutrophils in the central position [43]. Inflammatory edema follows the transmigration and activation of neutrophils [44]. The attenuated edema formation in group $B_{30h}$ suggests a relatively lower activation of neutrophils which, in turn, could explain the higher bacterial growth, supposedly associated with relatively lower bactericidal capacity. The edema formation correlates with the TNF-α responses at the bacterial challenge in the two groups. The present experiment was not a lung injury model, but $B_{30h}$ reached $PaO_2/FiO_2$ values bordering mild acute respiratory distress syndrome. However, $PaO_2/FiO_2$ decreased in $B_{30h}$ during the first 24 h despite intermittent recruitment maneuvers, and the lung parenchyma was significantly drier than in $C_{6h}$ at the end of the experiment. Most likely the lower $PaO_2/FiO_2$ in $B_{30h}$ reflected areas of atelectasis that existed at the time of bacterial challenge. Therefore, the anatomical conditions in the lungs differed between the groups from the start of the bacterial challenge. Another factor indicating atelectasis was that the $V_T$ was gradually reduced in $B_{30h}$ during the first 24 h and there was a significant difference between the groups throughout the bacterial phase. Atelectasis is a negative factor for bacterial clearance in porcine experimental settings. Thus, inversely, the relatively more open lung in $C_{6h}$ would be a favorable factor for reducing bacterial growth and the total bacterial load in lung tissue [31,45].

Our experiment has several limitations. Perhaps the most central factor for bacterial clearance was not evaluated in the experiment, i.e. the antibacterial capacity of the AM. Additionally, several possible influential co-factors discussed were not measured, namely surfactant, glycocalyx and complement. The experiment was not designed for these outcomes and would have needed a different laboratory set up but is in the plan for future experiments. Our experiment was arranged to resemble an intensive care setting with a minimal amount of inflammatory or potentially organ damaging insults to the research animals. With this configuration we actively chose to avoid repeated and large volume bronchoalveolar lavages after the bacterial inoculation for measurements and harvesting of alveolar macrophages and surfactant. Further,

we actively decided against the use of a bronchoscope and chose the minimally invasive method of blind bronchial sampling common to clinical practice. Interventions that potentially could have yielded different results in the BAL variables would probably have influenced the bacterial growth in lung tissue as well as inflammatory variables in the lungs. The BAL variables did not separate the groups in this experiment. Our efforts to enhance the results by additional analyses of urea dilution failed because of too many values below detection in BAL. The same problem was found when analyzing albumin in plasma and BAL for alveolo-capillary permeability measurements as a marker of ling injury. We believe the method of blind bronchial sampling used was too unspecific to yield better data. Lastly, there were no microscopic tissue comparisons regarding lung damage. Based on the different localities of sampling, the relatively few animals per group and problems of representability based on inhomogeneous lungs it was not prioritized in the experiment given that it was deemed an underpowered measurement. The translational relevance of the current experiment lies in that it uses clinically similar intensive care conditions and relevant equipment to evaluate different aspects and factors of influence on bacterial growth in the lungs. The results underline the importance of inflammatory state, i.e. the reactivity of the immune system to a bacterial challenge, for the clinical manifestations yielded by an infection. Additionally, the ease by which we unknowingly can affect immune system reactivity by anesthesia and intensive care calls for further exploration on the cellular and clinical level.

## Conclusions

Concomitant pre-exposure to MV and endotoxin increases the growth of *P. aeruginosa* in lung tissue from a subsequent bacterial challenge. Pre-exposure to only MV increases the bacterial growth similarly but to a lesser extent. In comparison, animals unexposed to MV or endotoxin show suppressed bacterial growth but at the cost of a more pronounced systemic inflammatory reaction with increased edema formation in lung tissue.

## Supporting information

**S1 Fig. Distributions of *Pseudomonas aeruginosa* growth.** Raw data presentations of bacterial cultures in the experiment based on a) left-right side distribution within each group (*post hoc* one-way ANOVA between left (L) and right (R) within each group, all $p > 0.05$), b) craniocaudal distribution regardless of group (*post hoc* one-way multiple ANOVA $p < 0.05$), mean indicated by line.
(TIF)

**S2 Fig. Distributions of wet-to-dry ratios.** Raw data presentations of all wet-to-dry measurements in the experiment based on a) left-right side distribution within each group (*post hoc* one-way ANOVA between left (L) and right (R) within each group, all $p > 0.05$), b) cranio-caudal distribution regardless of group (*post hoc* ANOVA p 0.23), mean indicated by line.
(TIF)

**S3 Fig. IL6 in plasma.** Mean±SEM, total group difference calculated between 0–6 h with ANOVA for repeated measures. Experimental parts *Inflammation* $A_{30h+Etx}$ vs. $B_{30h}$ (group p 0.84, group*time 0.49), *Ventilation Time* $B_{30h}$ vs. $C_{6h}$ (group p 0.63, group*time $p < 0.05^*$), axis scale changes at 0 hours. *Post hoc* statistics (group*time) present differing dynamics in cytokine escalation in $C_{6h}$ from the bacterial challenge 0–6 h.
(TIF)

**S1 Table. *Pseudomonas aeruginosa* and cytokines in bronchoalveolar lavages–complementary *post hoc* tests at– 24 and 0 h.** *Pseudomonas* (*P.*) *aeruginosa*, bronchoalveolar lavage

(BAL), colony forming unit (CFU), TNF-$\alpha$ (tumor necrosis factor alpha), IL6 (interleukin 6) at the start of the experiment -24 h ($A_{30h+Etx}$ and $B_{30h}$), at the bacterial inoculation 0 h and at the end of the experiment 6 h, median(LQ/HQ), p upper refers to $A_{30h+Etx}$ vs. $B_{30h}$, lower to $B_{30h}$ vs. $C_{6h}$.
(DOCX)

**S2 Table. Physiologic and laboratory variables in the sham animals.** Sham 30h ($S_{30h}$, n = 1), Sham 6h ($S_{6h}$, n = 2), heart rate (HR), mean arterial pressure (MAP), mean pulmonary arterial pressure (MPAP), cardiac index (CI), pulmonary capillary wedge pressure (PCWP), pressure (P), wet-to-dry ratio (WD), tumor necrosis factor alpha (TNF-$\alpha$), interleukin (IL). Sham 30 h ($S30$, n = 1) absolute value, sham 6 h ($S6$, n = 2), mean±SD, median (LQ/HQ) as presented in the main manuscript.
(DOCX)

**S1 Data.**
(XLSX)

**S1 File. Statement from statistician.**
(PDF)

**S2 File. Supplementary materials and methods.**
(DOCX)

## Acknowledgments

The authors would like to thank the staff at the Hedenstierna laboratory for excellent technical assistance; Lisa Maudsdotter at the University of Tsukuba for invaluable help with bacterial methods; Ulf Larsson, senior statistician and Andreas Pikwer at the Centre for Clinical Research Sörmland, Uppsala University, for their review of the statistical design and analyses. The study was performed at the Hedenstierna laboratory, Uppsala University Hospital, Uppsala, Sweden.

## Author Contributions

**Conceptualization:** Jesper Sperber, Axel Nyberg, Anders Krifors, Paul Skorup, Miklós Lipcsey, Markus Castegren.

**Formal analysis:** Jesper Sperber, Markus Castegren.

**Funding acquisition:** Markus Castegren.

**Investigation:** Jesper Sperber, Axel Nyberg, Anders Krifors, Paul Skorup, Markus Castegren.

**Methodology:** Jesper Sperber, Markus Castegren.

**Project administration:** Markus Castegren.

**Supervision:** Miklós Lipcsey, Markus Castegren.

**Validation:** Jesper Sperber.

**Writing – original draft:** Jesper Sperber.

**Writing – review & editing:** Jesper Sperber, Axel Nyberg, Anders Krifors, Paul Skorup, Miklós Lipcsey, Markus Castegren.

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
