## [Decision Letter · Decision Letter 0]

27 Jul 2020

PONE-D-20-18632

Pre-exposure to mechanical ventilation and endotoxemia additively increases Pseudomonas aeruginosa growth in lung tissue during experimental porcine pneumonia

PLOS ONE

Dear Dr. Sperber,

Thank you for submitting your manuscript to PLOS ONE. After careful consideration, we feel that it has merit but does not fully meet PLOS ONE’s publication criteria as it currently stands. Therefore, we invite you to submit a revised version of the manuscript that addresses the points raised during the review process.

Experimental work expanding our knowledge on lung physiology and pathology through studies into large animal models are highly appreciated. It is accepted that such studies are limited in the numbers of animals that can be included as well as in e.g. immunological parameters that can be analysed. However, as you may see from the detailed comments below, we would greatly appreciate if you could add some additional analyses from the BAL samples collected. Because you did not only collect BAL at the end of the experiment but also at -24h and 0h, respectively, TNF-alpha and IL-6 measurements at the other time points should be included. Such analyses should also comprise cell differentials to get an idea of the changes in immune cell composition in the alveolar space as well as additionally cytokines relevant at earlier time points of the pathologic processes in the lung. Data regarding BAL total protein should also be included. Further, the comments regarding the biostatistical analyses have to be considered. A detailed point-by-point response to the reviewers' comments will be mandatory.

We look forward to receiving your revised manuscript.

Kind regards,

Heinz Fehrenbach

Academic Editor

PLOS ONE

Additional Editor Comments:

I have a few additional comments expanding on what reviewer #1 (see below) stated.

- In the statistics part you state "Comparative group statistics in the experimental parts Inflammation (A30h+Etx vs. B30h) and Ventilation Time (B30hvs. C6h) were based on data from the last 6 h of the experiment (the bacterial phase). No multigroup comparisons including all three groups were used in the experiment." However, you used group B30h twice in your statitics, which means that you apparently treated the setting as two independent experiments. This is clearly not the case. Although being no biostatistician, I think you have to adjust the level of significanse to the fact that you used group B30h twice.

- In the conclusion part of the abstract you state: "Mechanical ventilation and systemic inflammation before the onset of pneumonia additively increase the growth of P. aeruginosa in lung tissue." I do not think that you are able to demonstrate additivity of the effects with your setting and I also cannot find a discussion of this aspect in the manuscript. Because the abstract should contain only well supported statements, I suggest that you remove this sentence or at least consderably modify it to comply with your results.

- Please translate all terms in the supplemnetal material to English language.

'The Animal Research Ethics Board of Uppsala issued the permit for the experiment (Uppsala djurförsöksetiska nämnd, DNr C 250/11).'    

(a) Please amend your current ethics statement to include whether the Animal Research Ethics Board of Uppsala specifically approved your study.  

(b) Once you have amended this/these statement(s) in the Methods section of the manuscript, please add the same text to the “Ethics Statement” field of the submission form (via “Edit Submission”).

For additional information about PLOS ONE ethical requirements for human subjects research, please refer to " ext-link-type="uri" xlink:type="simple">http://journals.plos.org/plosone/s/submission-guidelines#loc-human-subjects-research."

3. At this time, we request that you  please report additional details in your Methods section regarding animal care, as per our editorial guidelines:

(a) Please state the source and number of mice used in the study  

(b) Please provide details of animal welfare (e.g., shelter, food, water, environmental enrichment)

(c) Please provide the name and dosage of the specific anaesthetic agent used in your study

(d) Please include the method of euthanasia  

Thank you for your attention to these requests.

Reviewers' comments:

Reviewer's Responses to Questions

**Comments to the Author**

1. Is the manuscript technically sound, and do the data support the conclusions?

Reviewer #1: Partly

Reviewer #2: Yes

2. Has the statistical analysis been performed appropriately and rigorously? 

Reviewer #1: N/A

Reviewer #2: Yes

3. Have the authors made all data underlying the findings in their manuscript fully available?

Reviewer #1: Yes

Reviewer #2: Yes

4. Is the manuscript presented in an intelligible fashion and written in standard English?

Reviewer #1: Yes

Reviewer #2: Yes

5. Review Comments to the Author

Reviewer #1: Study by Sperber et al.

The study reports on the Pseudomonas aeruginosa growth of lung tissues after pre-exposure to mechanical ventilation with or without endotoxin-induced systemic inflammation in large animal (pig) models.

From interpretation point of view, there were three groups and two comparable parts in current study:

Group #1-A30h+Etx: endotoxin(+) + mechanical ventilation(+) + bacterial inoculation(+)

Group #2-B30h: endotoxin(-) + mechanical ventilation(+) + bacterial inoculation(+)

Group #3-C6h: endotoxin(-) + mechanical ventilation(-) + bacterial inoculation(+)

Comparable part #1-inflammation: A30h+Etx vs B30h

Comparable part #2-ventilation time: B30h vs C6h

The authors sampled lung tissues of three different regions (cranial, middle, and caudal) from both right and left lung of the same animal based on the heterogeneity of lung injury in large animals. Bacterial growth (colony forming unit) and lung edema (wet-to-dry ratio) from lung tissues are assessed as well as cytokines (TNF-α, IL 6 and 10) in bronchoalveolar lavage (BAL) and plasma to measure lung and systemic inflammation. The authors found significant bacterial growth of lung tissues in animals pre-exposed to mechanical ventilation and additionally to systemic inflammation for 24 hours. Lung edema was severe in A30h+Etx, and mild in B30h. The authors also found higher plasma TNF-α in C6h when compared to B30h. No differences of bacterial growth and cytokine levels in BAL between the groups were observed. This is an observational study on an important clinical topic performed in large animals. The methodology in this study is complete, and the statistical analysis appears appropriate. However, no mechanisms were investigated. Also, the causal relationship between each observation (bacterial growth, lung edema and systemic inflammation) seems not clear and needs to be improved.

Major comments:

1) From my understanding that the cytokines TNF-α and IL6 are the early responded factors during inflammation, it is anticipated that these cytokines would decrease or return to the normal levels later, as the results presented in current study. Thus, it's not clear why the authors measured TNF-α and IL6 at the end of current studies (the time periods were approximately 30 hours). Also, the higher levels of TNF-α in C6h might be due to the time frame after bacterial infection, in which time period TNF-α increased. Could the authors clarify these?

2) In terms of cytokines TNF-α and IL6 mainly produced by macrophages, these results in current studies might reflect the functional role for circulatory macrophages. How about the other immune cells, such as neutrophils, as they also play important roles in bacterial infection and clearance? It would be better if the authors could provide some evidence (such as other cytokines) to indicate the functional activity for other immune cells.

3) Due to the infection of P. aeruginosa in lung, I am curious about the local lung injury after pneumonia. In terms of lung damage, the authors measured oxygen index (PaO2/FiO2), wet-to-dry ratio, and cytokines in BAL. There was difference only in wet-to-dry ratio in each comparison. Since the lung tissues and BAL were harvested in current study, could the authors provide more measurements for lung damage, such as protein concentration in BAL, histological lung injury staining, cytokines and surfactants in lung tissues? These results would not only help support local lung injury after bacterial infection, but also might provide some potential explanation for bacterial growth.

4) In discussion of experimental part: A30h+Etx vs B30h, the authors discussed the potential effect of TNF-α peak and lung edema existed in A30h+Etx on impairing the immune system’s ability, which subsequently reduced bacterial growth. The results did not show difference in cytokines of BAL and plasma in A30h+Etx vs B30h at the end of study. Could the authors explain this? It would be more strength if the authors could provide more other evidences for the compromise immunity.

5) Page 26/Line 427-428: "Therefore, the higher TNF-α response to the bacterial challenge in C6h is an indication of the preserved functionality of the alveolar macrophages, ...". I think there would be "the higher plasma TNF-α response...". Could the authors improve the statement? How to explain the results of no difference of TNF-α in BAL between B30h and C6h, based on the preserved functionality of the alveolar macrophages in C6h?

6) Page26/Ling 438-441: "The attenuated edema formation in group B30 suggests a relatively lower activation of neutrophils which, in turn, could explain the higher bacterial growth, supposedly associated with relatively lower bactericidal capacity." In the discussion, the reason for the attenuated edema in B30h was attributed to the relatively lower activation of neutrophils. Could the authors provide the evidence for the lower activity of neutrophils in lung? This would enhance the strength of this statement and support the finding of higher bacterial growth.

Minor comments:

7) Page 9/Line 41 in Abstract: "A third group, C6h (n=8), ...". Please improve the description of the group C6h.

8) Page 9/Line 46-49: the results of abstract. Given there were two comparisons: A30h+Etx vs B30h and B30h vs C6h, it's not clear what comparison the P-values in results indicated. Could the authors describe these most important findings using numerical results (and statistical significance)?

9) Page 13/Line 140-142: "The alveolar recruitment maneuver (ARM) consisted of stepwise increments of PEEP until the peak pressure reached 35 cm H2O, followed by prolonged inspiration for 10 seconds (s)." Could the authors provide the details for the stepwise increments of PEEP (#-# cmH2O)?

10) Page 16/Line 216-220, the authors reported no difference of bacterial growth and wet-to-dry ratios between right and left sided lung. Could the authors improve the presentation of raw data from right and left side (supplementary Figure 1a and 2a) in each group, but not in all groups?

11) The P-value presented in Supplementary Figure 3 was a little bit confused based on only two comparable parts in this study. Could the authors clearly show the P values for each effect: inflammation, ventilation time and their interaction, in the figure legend of in the plot?

12) Page 21/Line 311: "The highest bacterial growth was in B30h (Figure 2)." Based on the comparison between B30h and C6h, perhaps it's better to present the data as: "The higher bacterial growth was in B30h...".

13) Page 26/Line 438-439: "The attenuated edema formation in group B30 suggests a relatively lower activation of neutrophils..." Please correct the group name "B30h".

14) Page 25/Line 411-412: "This result indicates a major edema development in

B30h+Etx only after the bacterial challenge." Here I think the group should be A30h+Etx.

Reviewer #2: The topic of this article is very interesting and relevant. I have just a few comments to make:

1) Although the hypothesis and research questions are well defined, were there any kind of sample size calculation

2) Please report data as mean and SD not as mean and sEM, especially since you have unequal group sizes.

3) I suggest tol elaborate a bit more the clinical / translational meaning of your findings.

4) Please comment why you did the inoculation in a blind way. Given the species it would have been easy to deliver everything in a very standardized way e.g. using fiberoptic bronchoscopy.

5) I suggest to explain also in this papers method section why you did not perform a sample size calculation. In my opinion crossreferencing is not sufficient in this case.

6. PLOS authors have the option to publish the peer review history of their article (what does this mean?). If published, this will include your full peer review and any attached files.

Reviewer #1: No

Reviewer #2: No

---

## [Author Response · Author response to Decision Letter 0]

8 Sep 2020

PONE-D-20-18632

Pre-exposure to mechanical ventilation and endotoxemia additively increases Pseudomonas aeruginosa growth in lung tissue during experimental porcine pneumonia

PLOS ONE

However, as you may see from the detailed comments below, we would greatly appreciate if you could add some additional analyses from the BAL samples collected. Because you did not only collect BAL at the end of the experiment but also at -24h and 0h, respectively, TNF-alpha and IL-6 measurements at the other time points should be included. Such analyses should also comprise cell differentials to get an idea of the changes in immune cell composition in the alveolar space as well as additionally cytokines relevant at earlier time points of the pathologic processes in the lung. Data regarding BAL total protein should also be included. 

Further, the comments regarding the biostatistical analyses have to be considered. A detailed point-by-point response to the reviewers' comments will be mandatory.

Please see further for responses to these comments.

Additional Editor Comments:

I have a few additional comments expanding on what reviewer #1 (see below) stated.

- In the statistics part you state "Comparative group statistics in the experimental parts Inflammation (A30h+Etx vs. B30h) and Ventilation Time (B30hvs. C6h) were based on data from the last 6 h of the experiment (the bacterial phase). No multigroup comparisons including all three groups were used in the experiment." However, you used group B30h twice in your statistics, which means that you apparently treated the setting as two independent experiments. This is clearly not the case. Although being no biostatistician, I think you have to adjust the level of significance to the fact that you used group B30h twice.

We do agree with the Editor that the questions regarding the statistical design are very central, and therefore, we have consulted an external senior statistician to critically revise our original approach and to comment. Please see original answer in supplementary file Statement from statistician.

In conclusion, the statistical review concluded that the objection from the editor could be divided into two parts. The two parts of the statistical concerns are commented below; 

1. Are the separate experiments independent since they share one group in common? Yes, the three individual groups should be considered independent and the two experiments can therefore be considered separate from a statistical viewpoint. 

2. Is a correction of the level of significance warranted? Correction is not warranted on the basis of the statistical status of the two separate experiments. However, correction could be warranted if the number of statistical analyses is high. As the investigation (both parts) is exploratory in nature with variables that biologically influence each other, no correction is deemed necessary. 

- In the conclusion part of the abstract you state: "Mechanical ventilation and systemic inflammation before the onset of pneumonia additively increase the growth of P. aeruginosa in lung tissue." I do not think that you are able to demonstrate additivity of the effects with your setting and I also cannot find a discussion of this aspect in the manuscript. Because the abstract should contain only well supported statements, I suggest that you remove this sentence or at least consderably modify it to comply with your results.

It is a valid point. Even if the two entities of mechanical ventilation and endotoxemia yield increasingly higher bacterial growth when added, compared to the unexposed animals, we have not proved it to be additively so. We have taken the word “additively” out of the title and the abstract.

- Please translate all terms in the supplemental material to English language.

The Swedish words in the Supplementary data file have been translated into English. 

'The Animal Research Ethics Board of Uppsala issued the permit for the experiment (Uppsala djurförsöksetiska nämnd, DNr C 250/11).' 

(a) Please amend your current ethics statement to include whether the Animal Research Ethics Board of Uppsala specifically approved your study. 

We have corrected the wording to approved and issued the permit for the current experiment (Ethical statement).

(b) Once you have amended this/these statement(s) in the Methods section of the manuscript, please add the same text to the “Ethics Statement” field of the submission form (via “Edit Submission”).

At this time, we request that you please report additional details in your Methods section regarding animal care, as per our editorial guidelines:

(a) Please state the source and number of mice used in the study 

The animals (Swedish farm pig) were acquired from a private source, Mångsbo Gård breeding facility, Uppsala, Sweden. In total 23 animals were used in the current experiment. (Added to the Ethics statement.)

(b) Please provide details of animal welfare (e.g., shelter, food, water, environmental enrichment)

The animals were allowed to eat and drink ad libitum up to 1 h before the start of the experiment. 

(c) Please provide the name and dosage of the specific anaesthetic agent used in your study

The pigs were sedated with tiletamin 3 milligrams (mg) x kilogram (kg)-1, zolazepam 3 mg x kg-1, and xylacin 2.2 mg x kg-1. Morphine 20 mg and ketamine 100 mg were given in an auricular vein. Anesthesia was maintained with pentobarbital 8 mg x kg-1 x h-1 and morphine 0.26 mg x kg-1 x h-1. To facilitate ventilator management and counteract shivering and coughing muscle relaxation was maintained with an infusion of rocuronium at an initial rate of 2 mg x kg-1 x h-1. 

(Added to the Ethics statement.)

(d) Please include the method of euthanasia 

Immediately after the experimental endpoint, the animals were euthanized by an intravenous injection of potassium chloride and mechanical ventilation was withdrawn.

(Added to the Ethics statement.)

Comments to the Author

5. Review Comments to the Author

Reviewer #1: Study by Sperber et al.

The study reports on the Pseudomonas aeruginosa growth of lung tissues after pre-exposure to mechanical ventilation with or without endotoxin-induced systemic inflammation in large animal (pig) models.

From interpretation point of view, there were three groups and two comparable parts in current study:

Group #1-A30h+Etx: endotoxin(+) + mechanical ventilation(+) + bacterial inoculation(+)

Group #2-B30h: endotoxin(-) + mechanical ventilation(+) + bacterial inoculation(+)

Group #3-C6h: endotoxin(-) + mechanical ventilation(-) + bacterial inoculation(+)

Comparable part #1-inflammation: A30h+Etx vs B30h

Comparable part #2-ventilation time: B30h vs C6h

The authors sampled lung tissues of three different regions (cranial, middle, and caudal) from both right and left lung of the same animal based on the heterogeneity of lung injury in large animals. Bacterial growth (colony forming unit) and lung edema (wet-to-dry ratio) from lung tissues are assessed as well as cytokines (TNF-α, IL 6 and 10) in bronchoalveolar lavage (BAL) and plasma to measure lung and systemic inflammation. The authors found significant bacterial growth of lung tissues in animals pre-exposed to mechanical ventilation and additionally to systemic inflammation for 24 hours. Lung edema was severe in A30h+Etx, and mild in B30h. The authors also found higher plasma TNF-α in C6h when compared to B30h. No differences of bacterial growth and cytokine levels in BAL between the groups were observed. This is an observational study on an important clinical topic performed in large animals. The methodology in this study is complete, and the statistical analysis appears appropriate. However, no mechanisms were investigated. Also, the causal relationship between each observation (bacterial growth, lung edema and systemic inflammation) seems not clear and needs to be improved.

Major comments:

1) From my understanding that the cytokines TNF-α and IL6 are the early responded factors during inflammation, it is anticipated that these cytokines would decrease or return to the normal levels later, as the results presented in current study. Thus, it's not clear why the authors measured TNF-α and IL6 at the end of current studies (the time periods were approximately 30 hours). Also, the higher levels of TNF-α in C6h might be due to the time frame after bacterial infection, in which time period TNF-α increased. Could the authors clarify these?

Thank you for a valuable comment. The TNF-α and IL6 responses are indeed early responses to inflammatory stimuli and will naturally wane to lower levels after an initial rise. However, the main point of the current experiment (two separate parts) was to relate bacterial growth in lung tissue to the inflammatory response during the bacterial phase which constituted the last six hours in all three groups. The inflammatory responsiveness at the time of an infection is clinically relevant for the patients’ ability to fight pathogens and for the effects of general inflammation, i.e. sepsis. Although these differences in clinical phenotype are known, they are hard to model in studies because of the problems of keeping experimental animals under long term intensive care before experimental interventions. Our experiment is an effort in this direction.

We constructed a way to investigate the influence of inflammatory responsiveness on bacterial growth. It was our hypothesis that prior inflammatory activation as seen from endotoxin infusion would entail a lower response in TNF-α and IL6 when given a secondary inflammatory challenge in the form of bacteria. Further we hypothesized that the secondary inflammatory response would affect the bacterial growth. It was not anticipated that the same response would be seen in the animals that did not receive endotoxin during 24 h. The discussion about the phenomenon of endotoxin tolerance, what sets it in play and its influence on bacterial growth is directed at this experimental layout. The cytokine response in the 6 h group, that did not have any prior inflammatory insult except preparatory surgery, represent the unaffected inflammatory system. Indeed, the cytokine response is higher in the 6 h group and entails both lower bacterial levels and higher edema formation in the lungs. We strongly believe that the time frame has influence on the cytokine responses. The investigation of the influence of time in mechanical ventilation was the intention of the experimental part named Ventilation Time, in which we compared one 30 h group with the 6 h group. 

To clarify we have added a sentence in the Statistics section (p 10, line 216).

2) In terms of cytokines TNF-α and IL6 mainly produced by macrophages, these results in current studies might reflect the functional role for circulatory macrophages. How about the other immune cells, such as neutrophils, as they also play important roles in bacterial infection and clearance? It would be better if the authors could provide some evidence (such as other cytokines) to indicate the functional activity for other immune cells.

Thank you for a valid comment. Macrophages are described as the main source of inflammatory cytokines in the initial response to an invading pathogen. Circulating leukocytes, as opposed to tissue macrophages, have proven to not be a major source of cytokines in human sepsis as described by Gille-Johnson et al 2012. In the lungs, the alveolar macrophage is abundant in numbers and a very likely major contributor to systemic cytokines in the current experiment. Additional sources of cytokines are recruited neutrophils and endothelia. These cells all have in common an ability to produce nitric oxide (NO) by inducible nitrous oxide synthase. The NO molecule has many disparate effects in an infection, not least to act bactericidal and produce capillary leakage. To target immune cell function better we made additional analyzes of nitrite in urine as a proxy for total NO turnover. The NO molecule is highly unstable and short lived why this stable byproduct is more practical to analyze. The results show that both 30 h groups had suppressed levels of nitrite to around 20% of baseline, and that the 6 h group had significantly higher levels during the whole bacterial phase. We interpret this as a mechanistic explanation of our findings. Exemplified by the 6 h group – higher nitrite indicates higher immune cell activity which entails lower bacterial levels and higher edema development in the Ventilation Time experiment. We have made additions to Results, added a new Figure (5) and comments in the discussion. (p 20, line 437)

3) Due to the infection of P. aeruginosa in lung, I am curious about the local lung injury after pneumonia. In terms of lung damage, the authors measured oxygen index (PaO2/FiO2), wet-to-dry ratio, and cytokines in BAL. There was difference only in wet-to-dry ratio in each comparison. Since the lung tissues and BAL were harvested in current study, could the authors provide more measurements for lung damage, such as protein concentration in BAL, histological lung injury staining, cytokines and surfactants in lung tissues? These results would not only help support local lung injury after bacterial infection, but also might provide some potential explanation for bacterial growth.

Thank you for a valuable comment. During the planning of the experiment we considered alternative BAL methods. However, since the experiment entailed pulmonary outcomes, and the development of edema and P/F ratios were relevant physiological outcome variables we decided to use as little intervention as possible within the lungs. Our choice of a minimal procedure, as is common in clinical practice, was therefore chosen deliberately for this experiment. Repeated BALs or higher volume BAL would have affected the model for our primary outcomes negatively, although the BAL itself may have had higher quality. 

As an additional effort to improve the BAL results, we tried to calculate the dilution factor of the BAL sample to normalize each sample and make more effective comparisons. The basis for this approach was that different samples may have had different amount of saline added to the alveolar lining fluid and standardization could impact the results. The dilution factor could potentially be used on both cytokines and bacterial counts in BAL. We analyzed urea in plasma and BAL, a method described for the purpose, as urea supposedly freely flows to the alveolar lining fluid yielding the same concentration as in plasma. Regrettably, most urea values were below detection limit in the BAL fluid effectively making this approach not viable. Our conclusion is that the dilution from saline in the BAL fluid was substantial but not quantifiable in the current experiment.

Regarding lung injury and protein counts as suggested from the editor and reviewer, we have made additional analyzes of albumin in plasma and BAL to evaluate the alveolo-capillary permeability. We chose to use an albumin rather than a total protein assay as the alveolo-capillary leakage would be influenced by the size of the protein studied. A total protein assay would detect proteins of different molecular sizes while an albumin method only measures albumin with a molecular weight of 67 kDa. Another problem with total protein methods is that the methods do not measure all proteins equally. E.g. 1 mg bovine serum albumin gives approximately twice the absorbance as 1 mg IgG with an assay utilizing a Coomassie reagent. Like urea most albumin samples were below detection limit in BAL fluid and the variable could not be produced. As a total protein analysis would suffer from the same problem of dilution as the other variables in BAL fluid it was decided a non-viable endeavor and not pursued further. 

Regarding tissue samples we did only structurally harvest samples that were used for cultures and wet-to-dry weight measurements. We initially considered to save samples for later tissue analysis but decided against it as we were unsure about where the samples should be harvested to get representability since the lungs where highly inhomogeneous. Therefore, we don’t have samples to analyze that would benefit the experiment. 

Summarily, the method of BAL did not yield data of sufficient quality to make conclusions in the current experiment. Efforts to better the results in our experiment by additional analyses regrettably did not deliver usable data. The additional analyzes, urea and albumin, are referred to in the main manuscript Results section but data is only presented in the Supplementary data file as they were not fit for tabular presentation. An updated comment on the BAL method is added to the Discussion (p 23, line 514).

4) In discussion of experimental part: A30h+Etx vs B30h, the authors discussed the potential effect of TNF-α peak and lung edema existed in A30h+Etx on impairing the immune system’s ability, which subsequently reduced bacterial growth. The results did not show difference in cytokines of BAL and plasma in A30h+Etx vs B30h at the end of study. Could the authors explain this? It would be more strength if the authors could provide more other evidences for the compromise immunity.

We believe the BAL method was inadequate to produce better quality results. Our efforts to improve the quality of the BAL samples by additional analyses of urea regrettably failed (see answer to comment 3). 

The analysis of nitrate in urine addresses the question of additional data regarding compromised immune function (see answer to comment 2).

5) Page 26/Line 427-428: "Therefore, the higher TNF-α response to the bacterial challenge in C6h is an indication of the preserved functionality of the alveolar macrophages, ...". I think there would be "the higher plasma TNF-α response...". Could the authors improve the statement? How to explain the results of no difference of TNF-α in BAL between B30h and C6h, based on the preserved functionality of the alveolar macrophages in C6h?

The sentence is corrected accordingly. The lack of difference in BAL cytokines is addressed in the previous comment. The preserved functionality of alveolar macrophages (and other main cytokine producers such as neutrophils and endothelia) in the 6 h group is depicted in higher levels of pro-inflammatory cytokines in plasma and in higher levels of urinary nitrate (see answer to comment 2).

6) Page26/Ling 438-441: "The attenuated edema formation in group B30 suggests a relatively lower activation of neutrophils which, in turn, could explain the higher bacterial growth, supposedly associated with relatively lower bactericidal capacity." In the discussion, the reason for the attenuated edema in B30h was attributed to the relatively lower activation of neutrophils. Could the authors provide the evidence for the lower activity of neutrophils in lung? This would enhance the strength of this statement and support the finding of higher bacterial growth.

The additional results from nitrite analyses in urine comprises the total function of main producers of NO from iNOS. We cannot separate the part attributable to neutrophils. B30h had approximately 20% of the nitrite level of C6h at the time of bacterial challenge, and the difference was significant during the bacterial phase. We have no further possibility to analyze neutrophil function, regrettably.

Minor comments:

7) Page 9/Line 41 in Abstract: "A third group, C6h (n=8), ...". Please improve the description of the group C6h.

The description is corrected into “A third group, C6h (n=8), started the experiment at the bacterial inoculation unexposed to endotoxin or mechanical ventilation (total experimental time 6 h)”.

8) Page 9/Line 46-49: the results of abstract. Given there were two comparisons: A30h+Etx vs B30h and B30h vs C6h, it's not clear what comparison the P-values in results indicated. Could the authors describe these most important findings using numerical results (and statistical significance)?

The abstract is updated and corrected for greater clarity.

9) Page 13/Line 140-142: "The alveolar recruitment maneuver (ARM) consisted of stepwise increments of PEEP until the peak pressure reached 35 cm H2O, followed by prolonged inspiration for 10 seconds (s)." Could the authors provide the details for the stepwise increments of PEEP (#-# cmH2O)?

The sentence has been corrected into “The alveolar recruitment maneuver (ARM) consisted of stepwise increments of PEEP (sequential 3 cm H2O increments for 5 seconds (s) each under control of systolic arterial blood pressure) until the peak pressure reached 35 cm H2O, followed by prolonged inspiration for 10 s.

10) Page 16/Line 216-220, the authors reported no difference of bacterial growth and wet-to-dry ratios between right and left sided lung. Could the authors improve the presentation of raw data from right and left side (supplementary Figure 1a and 2a) in each group, but not in all groups?

The Supplemental Figures 1a and 2a are updated accordingly and the captions are corrected.

11) The P-value presented in Supplementary Figure 3 was a little bit confused based on only two comparable parts in this study. Could the authors clearly show the P values for each effect: inflammation, ventilation time and their interaction, in the figure legend of in the plot?

The primary statistics is presented in Table 4 Plasma cytokines, inflammatory cells and temperature and describes only p-values for the group factor (Inflammation p 0.84, Ventilation Time p 0.63) and not the interaction of group*time. No differences between groups are present based on the group factor only. Group*time can indicate a different dynamic between two comparable groups even if the total sum of values is similar – such as the case with IL6. As this dynamic is of some interest especially in C6h and it is mentioned briefly in the manuscript, we chose to add a presentation as a supplementary file. The caption is updated with additional information.

12) Page 21/Line 311: "The highest bacterial growth was in B30h (Figure 2)." Based on the comparison between B30h and C6h, perhaps it's better to present the data as: "The higher bacterial growth was in B30h...".

The point is valid, and the sentence is changed accordingly.

13) Page 26/Line 438-439: "The attenuated edema formation in group B30 suggests a relatively lower activation of neutrophils..." Please correct the group name "B30h".

Corrected accordingly.

14) Page 25/Line 411-412: "This result indicates a major edema development in

B30h+Etx only after the bacterial challenge." Here I think the group should be A30h+Etx.

Corrected accordingly.

Reviewer #2: The topic of this article is very interesting and relevant. I have just a few comments to make:

1) Although the hypothesis and research questions are well defined, were there any kind of sample size calculation

No power calculation was conducted for this specific experiment since we had no previous data on bacterial behavior in our models. Instead, we used the power calculation for the preceding inflammatory experiments (referred to as 6-8 in the main manuscript). It was based on a systemic TNF-alpha difference of 15% at 6 hours, an alpha error of 0,05, a power of 0,8, and an SD of 10%, which yielded six evaluable animals per group. The choice of 8 animals per group in the previously published day-based experiment (8) was based on this calculation while allowing for a slightly larger variability in the bacterial outcome variable. 

As we started with the daybased (8) experiment we could appreciate the bacterial growth in lung tissue better. Based on this data we reduced the number of animals in the 30 h experiments, which were completed at the end of the experimental period, from eight to six to meet the 3R principle. In summary, we reduced the number of animals as we believed we could meet the required difference in the main outcome variable anyway.

The above sequence is added to the end of the Statistics section in the manuscript (p 11, line 243).

2) Please report data as mean and SD not as mean and sEM, especially since you have unequal group sizes.

All tabular data in the manuscript is presented as mean±SD or median(LQ/HQ) as appropriate. In the graphical presentations we find that mean±SEM (66.6% confidence interval) better presents the message intended graphically. One figure, nitrate in urine, is presented as a non-parametric box plot with median(LQ/HQ). The rational for tabular and graphical presentations have been the standard in our previous publications. Importantly, the comparative statistics is not influenced by these choices of presentation.

3) I suggest to elaborate a bit more the clinical / translational meaning of your findings.

We have made additions to the sequence in the manuscript.

The translational relevance of the current experiment lies in that it uses clinically similar intensive care conditions and relevant equipment to evaluate different aspects and factors of influence on bacterial growth in the lungs. The results underline the importance of inflammatory state, i.e. the reactivity of the immune system to a bacterial challenge, for the clinical manifestations yielded by an infection. Additionally, the ease by which we in clinical care unknowingly can affect immune system reactivity by anesthesia and intensive care calls for further exploration on the cellular level. (p 23, line 529)

4) Please comment why you did the inoculation in a blind way. Given the species it would have been easy to deliver everything in a very standardized way e.g. using fiberoptic bronchoscopy.

We agree with the view that a more thorough BAL method potentially could have given different results in the BAL variables. They were not the primary outcome variables in the current experiment and we actively decided to use a minimally invasive method so as to affect the lungs as little as possible. This is partly addressed in a previous comment from reviewer 1 (comment 3), and a sentence is added in the discussion on limitations (p 23, line 516) .

5) I suggest to explain also in this papers method section why you did not perform a sample size calculation. In my opinion crossreferencing is not sufficient in this case.

The explanation for group sizes is added to the statistics section, see response to comment 1.

---

## [Decision Letter · Decision Letter 1]

2 Oct 2020

Pre-exposure to mechanical ventilation and endotoxemia increases Pseudomonas aeruginosa growth in lung tissue during experimental porcine pneumonia

PONE-D-20-18632R1

Dear Dr. Sperber,

We’re pleased to inform you that your manuscript has been judged scientifically suitable for publication and will be formally accepted for publication once it meets all outstanding technical requirements.

Kind regards,

Heinz Fehrenbach

Academic Editor

PLOS ONE

Additional Editor Comments (optional):

All my specific comments were adequately addressed.

Reviewers' comments:

Reviewer's Responses to Questions

**Comments to the Author**

1. If the authors have adequately addressed your comments raised in a previous round of review and you feel that this manuscript is now acceptable for publication, you may indicate that here to bypass the “Comments to the Author” section, enter your conflict of interest statement in the “Confidential to Editor” section, and submit your "Accept" recommendation.

Reviewer #1: All comments have been addressed

Reviewer #2: All comments have been addressed

2. Is the manuscript technically sound, and do the data support the conclusions?

Reviewer #1: Yes

Reviewer #2: Yes

3. Has the statistical analysis been performed appropriately and rigorously? 

Reviewer #1: Yes

Reviewer #2: Yes

4. Have the authors made all data underlying the findings in their manuscript fully available?

Reviewer #1: Yes

Reviewer #2: Yes

5. Is the manuscript presented in an intelligible fashion and written in standard English?

Reviewer #1: Yes

Reviewer #2: Yes

6. Review Comments to the Author

Reviewer #1: Thank you for addressing all my concerns to great detail. Authors also performed additional work. The manuscript has significantly improved. I have no additional questions.

Reviewer #2: (No Response)

7. PLOS authors have the option to publish the peer review history of their article (what does this mean?). If published, this will include your full peer review and any attached files.

Reviewer #1: No

Reviewer #2: No

---

## [Editor Report · Acceptance letter]

15 Oct 2020

PONE-D-20-18632R1 

Pre-exposure to mechanical ventilation and endotoxemia increases *Pseudomonas aeruginosa* growth in lung tissue during experimental porcine pneumonia 

Dear Dr. Sperber:

I'm pleased to inform you that your manuscript has been deemed suitable for publication in PLOS ONE. Congratulations! Your manuscript is now with our production department. 

Kind regards, 

on behalf of

Prof. Dr. Heinz Fehrenbach 

Academic Editor

PLOS ONE